# Beyond Patient Characteristics: A Narrative Review of Contextual Factors Influencing Involuntary Admissions in Mental Health Care

**DOI:** 10.3390/healthcare11141986

**Published:** 2023-07-09

**Authors:** Deborah Oyine Aluh, Osaro Aigbogun, Blessing Onyinye Ukoha-Kalu, Manuela Silva, Ugnė Grigaitė, Barbara Pedrosa, Margarida Santos-Dias, Graça Cardoso, José Miguel Caldas-de-Almeida

**Affiliations:** 1Lisbon Institute of Global Mental Health (LIGMH), NOVA Medical School, NOVA University of Lisbon, 1150-082 Lisboa, Portugal; 2Comprehensive Health Research Center (CHRC), NOVA Medical School, NOVA University of Lisbon, 1150-082 Lisboa, Portugal; 3Department of Clinical Pharmacy and Pharmacy Management, University of Nigeria Nsukka, Nsukka 410105, Nigeria; 4Department of Management, Marketing and Digital Business, Curtin University, Miri 98009, Malaysia; 5School of Medicine, University of Nottingham, Nottingham NG7 2RD, UK

**Keywords:** involuntary admission, context, legislation, service, staff, coercion, public attitudes

## Abstract

Variations in the rates of involuntary admission (IA) reflect the influence of unexplained contextual variables that are typically too heterogeneous to be included in systematic reviews. This paper attempts to gather and analyze factors unrelated to the patients that have been linked to IA. The articles included in this review were selected by iteratively searching four electronic databases (PubMed, PsychINFO, EMBASE, and Web of Science). A total of 54 studies from 19 different countries and regions, including 14 European countries, the United States, Canada, China, Vietnam, and Taiwan, were selected. The factors were categorized as service-related factors, impactful events, seasonal and temporal factors, mental health legislation, staff factors, and public attitudes. The factors rarely act in isolation but rather interact and reinforce each other, causing a greater influence on IA. This paper explains how these factors present opportunities for robust and sustainable interventions to reduce IAs. The paper also identifies future directions for research, such as examining the effects of economic recessions. Enhancing global reporting standards is essential to validate future research and support further in-depth studies. The complexity of the factors influencing IA and the implicit role of society suggest that resolving it will require social change.

## 1. Introduction

Coercive measures in mental health care, including involuntary admission (IA), are a significant infringement of human rights and autonomy [1]. Their effectiveness for treating people with mental health conditions (PMHCs) is debatable [2,3], and they can have negative effects on therapeutic alliance, quality of life, and self-esteem [4,5]. Involuntary admission has been criticized as a violation of international human rights treaties [6], and those who experience it are more likely to experience shame, self-stigma, and poorer recovery outcomes [7]. However, in cases of severe danger to self or others, involuntary admission may be justified [8,9] and associated with improved psychosocial functioning and better motivation for treatment in some patients [10]. 

Although IAs are controversial and their effectiveness remains unclear because of the ethical problems associated with their investigation, they seem to be on the rise and deeply rooted in routine mental health care [11,12]. Wide variations in IA rates have been reported, ranging from 282 per 100,000 individuals in Austria to 14.5 per 100,000 individuals in Italy, with higher rates of IA associated with a higher number of inpatient beds, higher GDP per capita purchasing power parity, lower levels of absolute poverty, and a higher proportion of immigrants [13]. These variations have also been recorded among different regions within the same country [14,15]. Variations in IA rates among hospitals within the same jurisdiction were reported in Ireland, and the differences in admission status between patients remained unexplained [8]. These differences at different levels suggest that IA may not necessarily be employed as a last resort or in fulfilment of criteria specified in the mental health legislation [16], and that contextual factors may have a greater impact on IA than is often recognized.

Over the past two decades, there has been exponential growth in the literature on IAs and its contributing factors. Evidence syntheses from several studies globally have shown a socio-demographic patterning of IA, with gender, ethnicity, and level of social support consistently appearing to be universal factors associated with an increased risk of being involuntarily admitted [17,18]. Clinical factors such as lack of insight, a diagnosis of psychosis, and poor functioning have also shown an association with a higher risk of IA [19,20,21]. Most previous reviews about IA have focused on socio-demographic and clinical factors, usually with strict restrictive criteria that prevent exploring other potentially relevant variables. Despite the growing number of studies exploring non-patient related factors associated with IA, there has been no concerted effort to summarize and evaluate the findings from these studies. This review aims to fill this gap by including observational studies to provide a broad overview of factors, beyond those related to the patient, that could influence IA. Better information regarding contextual factors that influence IA is important for theoretical and practical reasons. From a theoretical standpoint, gaining an understanding of these factors holds implications for fully comprehending the problem of IA, particularly considering the predominant emphasis placed on individual risk factors associated with patients. On a practical level, it is vital for healthcare providers and service managers to grasp these factors, which often lie beyond their sphere of control, in order to effectively plan and adapt their services as needed. Moreover, the identification of these factors and the recognition of this problem within the broader context of underinvestment in mental health systems are significant concerns that policymakers should be interested in. Finally, our discussion should assist future researchers in their attempt to conduct studies that will provide deeper insights into the ambiguous aspects of contextual factors impacting IA. Patient factors are not discussed in this paper because they have already been adequately explored in previous reviews [17,18]. The objective of this paper is to identify and synthesize the literature on non-patient-related factors that have the potential to influence IA. 

## 2. Methods

We adopted a narrative approach for this review, which proved highly valuable for this study as it encompassed not only the identification of the factors that influence IA within the existing literature, but also enabled the synthesis of these factors by organizing them into broader themes. The SANRA scale for the evaluation of the quality of narrative review articles informed the methodological approach of this narrative review [22]. 

### 2.1. Literature Search

In healthcare literature, contextual factors are often viewed as barriers or facilitators to implementing an intervention [23]. For the purpose of this review, we define contextual factors as diverse elements within a specific setting that are unrelated to the patients themselves but that can exert a significant influence on IA directly or indirectly, while IA refers to compulsory admission under a mental health act. We hypothesized that the prevailing mental health laws, the quality of mental health services available, and stigma were factors that could significantly impact IA, irrespective of the patient’s clinical and socio-demographic profile. Four accredited electronic databases (PubMed, PsychINFO, EMBASE and Web of Science) were iteratively searched between 20 July 2022 and 11 August 2022 using specific search strings, first alone and then in combination. Search terms used include “involuntary admission”, “mental health services”, “staff”, “prevalence”, “predictors”, “context”, “socio-legal”, “socio-economic”, “public attitudes”, “legislation”, “attitudes”, and “factors” in various combinations. Database queries with an overlap of variants and synonyms of the search terms were adapted for each database in accordance with the appropriate request vocabulary. Boolean operators and wildcards were used to maximize the search performance. A manual search was further conducted by backward and forward citation tracking. All observational studies that investigated the association between any contextual factor and IA and were published in peer-reviewed journals since the inception of each database were considered. A total of 8660 articles were obtained from the four databases and uploaded into Rayyan software [9]. 

### 2.2. Selection of Studies for Review

A pilot screening of article titles and abstracts was performed by two authors (DOA and OA) using preliminary inclusion criteria. Careful deliberations after the pilot screening led to the establishment of a refined set of criteria for inclusion and exclusion (Table 1). Studies that explored the factors influencing the decision to commit were included. When a study included patient-related factors as well as a contextual factor of interest, only the contextual factor was considered. We chose to exclude qualitative studies because the majority of the qualitative studies that assessed contextual factors were related to the implementation of an intervention or the subjective opinions of professionals, service users, and their families. Studies conducted in specialized settings or among specific populations were also excluded because they often have distinct regulations, funding, outcomes, and ethical concerns that need analysis using different lenses; it would therefore exceed the scope of this review to include these settings. Although no research specifically investigated the direct link between public attitudes towards IA and IA rates, we recognized that this aspect is frequently disregarded despite its substantial influence on other contextual factors. Consequently, we chose to incorporate the existing studies that have examined public opinions on IA, even if they do not explicitly demonstrate their impact on IA rates. A PRISMA Flow Diagram illustrates the steps taken during the literature search and the number of articles included at each step (Figure 1).

### 2.3. Data Abstraction and Analysis

A total of 54 articles were selected and read carefully. Relevant data were extracted into a table that enabled the contents of each article to be summarized (see Appendix A). The extant literature on contextual factors that influence IA is diverse and fragmented, so we attempted to organize and integrate the selected studies. We employed a combination of deductive reasoning, utilizing factors we hypothesized, and inductive reasoning, based on our analysis of the selected papers, to establish categories of contextual factors. These categories served as a framework for organizing the papers accordingly. They include factors related to the services, factors related to seasonal and temporal variations, the impact of significant events, the influence of mental health laws, factors related to staff, and public attitudes. The results section was organized according to these. The review was intended to encompass a wide range of literature on the subject, and no quality rating of the articles was performed as it is considered optional for narrative reviews [22].

## 3. Results

A total of 54 full texts articles were included in this review. The studies were from 19 different countries and regions, including 14 European countries, the United States, Canada, China, Vietnam, and Taiwan. A summary table with information about the selected studies can be found in the Appendix A.

### 3.1. Service-Related Factors

Sixteen articles that explored factors related to services that might affect IA were identified. These factors include the level of development of alternative services to hospitalization, the efficiency and extent of utilization of existing mental health services, the availability of supplementary social services, and the number and location of psychiatric hospital beds. Since patients who have received adequate care are less likely to experience crises requiring admission, IA can be an indicator of the quality of prior care received [24,25]. The closure of large stand-alone psychiatric hospitals with the concurrent development of psychiatric departments in general hospitals and community-based mental health services is promoted as the ideal model of service organization in mental health care since it encourages continuity of care, higher user satisfaction, better protection of human rights, and reduces stigma [26]. The lack of effective and less restrictive alternatives to inpatient treatment is frequently cited as a reason for IA [27]. 

#### 3.1.1. Availability of Community-Based Services

Few studies have specifically assessed the impact of the availability of alternative services, such as intensive community programs, on IA. In a study to determine the effect of the development of alternative services in French psychiatric sectors on involuntary inpatient care, the likelihood of a patient being forced into full-time hospitalization decreased by 12% for every 10% increase in the level of development of alternatives [28]. The same study also found that the number of community-based psychiatrists in each catchment area significantly influenced the rates of IA [28]. In Denmark, an observational study that specifically examined the effect of assertive community treatment (ACT) on the use of several coercive measures, including IA, found that while the reduction in voluntary admissions was significant, there was no corresponding pattern in IA. There was a nonsignificant drop in IA due to the danger criterion, but no reduction in the other forms of IA. Admissions based on the treatment criterion occurred at the same rate before and after the use of ACT [29]. According to a US study, the location of the assessment of a person undergoing a mental health crisis seems to influence the decision to admit involuntarily. IA was initiated three times more frequently when clients were assessed in hospitals and police stations as opposed to other community venues, like the client’s home [30]. This indicates that if people experiencing a mental health crisis could be referred to intensive community programs with trained staff rather than hospitals, the rate of IA could decrease [30]. 

#### 3.1.2. Efficiency of Mental Health Services

The availability of mental health services is apparently not the most important aspect of mental health systems but rather how well the services perform. Bindman et al.’s study in England on the effects of social deprivation and the functioning of local mental health services on IA discovered significant associations between measures of service function at a sector level and rates of IA, which persisted even when deprivation was considered [31]. Teams that provided home visits to critically ill patients after 10 p.m. held fewer people for assessment [31]. An earlier study in England and Wales found that locations with insufficient funding, and poorly coordinated emergency services had an increased risk of high rates of emergency IA [32]. These studies suggest that fewer IA and high-quality aftercare can be expected in areas where services have formal or collaborative relationships. Another factor impacting the risk of IA is the extent of utilization of the available mental health services. In an Amsterdam study of acute psychiatry, more intensive psychiatric care received before an emergency consultation was significantly associated with a 70% lower risk of IA. A history of more than 14 outpatient contacts in the previous year was also associated with a lower risk of IA [33]. In the same study, a referral from a general practitioner was associated with a reduced probability of an IA than referrals from the police or mental health services, highlighting the importance of working together with primary care physicians to lower the number of IAs. In another study conducted in Greece, individuals who had contact with community mental health services before being admitted had a six times lower risk of being subjected to IA than those whose previous contact with mental health providers had been through a previous hospitalization [34]. 

#### 3.1.3. Availability of Complementary Social Services

Social services that complement the services provided by psychiatric services are necessary to meet the non-clinical needs of PMHCs. The number and variety of social services available in a given area may influence the risk of IA. An investigation into how these services could affect the voluntariness of admission in 13 German adult psychiatric hospitals found a correlation between the activities of social-psychiatric services, community-psychiatric cooperation, and complementary facilities and the IA index. Low levels of social-psychiatric services were found in regions with a high IA index, indicating that those services had a protective effect on acute psychiatry [35]. Another German study discovered that clinics with low rates of IA were more community-oriented and had more crisis intervention resources, such as home visits, close relationships, and low-threshold contact services, demonstrating that the range and quality of local social-psychiatric services had a significant influence on the rates of IA [36]. The number of social workers in a mental health service was found to be a significant contextual factor influencing the rate of IA in an Italian study where IA was better predicted by the team structure and service organization than by patient characteristics. IAs rose as the number of social workers in the departments increased [37]. The authors proposed a potential interpretation of this discovery, suggesting that the significant presence of social workers within the services resulted in an increased ability to identify crises situations within the community and exercise greater social control.

#### 3.1.4. The Number and Location of Psychiatric Hospital Beds 

The association between the number of psychiatric hospital beds and the rates of IA is unclear, with some research finding no association and others finding considerable effects. An international comparison of IA national rates among 22 countries across Europe, Australia, and New Zealand found no association with the number of beds [13]. In Italy, even though the total number of psychiatric beds decreased by 62.5%, over the 18 years following the implementation of the new mental health law, the proportion of IA decreased from 17.1% to 10.6% [38]. Of course, the reduction of psychiatric beds was not the only element at play in this situation because community-based services were also growing at the same time that the mental health legislation was changing. In contrast to the findings in Italy, an ecological study in England of the relationship between the availability of beds for people with mental illnesses and the rate of involuntary admissions in the NHS found that between 1988 and 2008, the rate of involuntary admissions increased by more than 60% while the availability of beds decreased by more than 60%, with the changes appearing to be synchronous [39]. Smith et al., also discovered some evidence that cutbacks in mental health bed availability contributed to some of the post-2008 increases in IA and Place of Safety detentions in England [40]. The decreased number of beds available amplified the impact of austerity measures implemented during the economic recession on IA [40]. This demonstrates how reducing psychiatric beds without concomitant investment in community resources might result in unfavorable effects such as increasing IAs. In another study conducted in Virginia, US, it was discovered that when detention beds were available, patients had significantly higher risks of being detained [41]. Having psychiatric beds close by seemed to reduce the use of IA in a Norwegian study that looked at how several factors, including the organization of mental health facilities, affected rates of compulsory admission. They found that the deinstitutionalized system with no local beds available had the highest levels of IA. For the systems with local beds, increased accessibility, local control, and integration of psychiatric beds and services lowered the threshold for inpatient admission, enabling patients to be admitted voluntarily more easily before their condition worsened [42]. Conversely, in a comparative case register study of seven Nordic catchment areas, Hansson et al., found that lower procedural accessibility of psychiatric specialist services, in terms of the need for a formal referral procedure, was associated with higher rates of compulsory care [43]. Very few effects were found in a study that assessed the effect of service integration (defined as a variety of actions, such as reorganization, new support and consultation structures, and information system regrouping) on IA. Variables other than service integration appeared to be more significant in lowering IA, but service outcomes were generally better where mental health care was more integrated [44]. 

### 3.2. Temporal and Seasonal Factors

Eight papers investigated the relationships between IA and temporal and seasonal parameters. 

#### 3.2.1. Temporal Factors

A multi-center investigation of twenty acute psychiatric units in Norway found that IA happened more often during evenings and nights and by physicians who did not know the PMHC [45]. Two separate Swiss studies reported that involuntary admissions occurred more often at night time and during the weekends [46,47]. The link between “daytime hours” and fewer IAs was attributed to a combination of demographics, clinical features, and out-of-clinic service availability (such as ambulatory psychiatric–psychological praxis; day-clinic; and home treatment) [46]. A German study likewise discovered that cases admitted unwillingly were admitted more frequently at night or on weekends [36]. These studies emphasize the significance of putting in place after-hours outpatient crisis intervention programs. In contrast to this finding, a UK study of consecutive psychiatric hospital admissions found that IA was 15% less frequent on the weekend than it was during the week after correcting for other demographic and clinical characteristics [48]. This was attributed to a shortage of required mental healthcare professionals and auxiliary personnel to implement IA throughout the weekend. The study findings were thus reflective of the poor availability of primary care, community mental health, and social care services on weekends [48]. This pattern is not exclusive to European countries as a Taiwanese study also found that patients admitted on weekends tended to be compulsorily admitted [49].

#### 3.2.2. Seasonal Factors

A thorough investigation of the environmental factors associated with involuntary hospitalizations could help predict and prevent high levels of psychiatric emergencies leading to IA. Few studies have specifically explored the impact of environmental factors on IA. The admissions of patients with schizophrenia and bipolar disorder have been found to vary seasonally following variations in sunshine intensity [50,51]. These variations have been linked to irregular chronobiology, which causes insomnia and dysregulation of circadian rhythms in those who have bipolar disorder [52] and could also be influenced by photoperiod changes, which cause symptoms to deteriorate and cause exacerbations in those with schizophrenia [51]. An Italian study found a considerable rise in involuntary hospitalization with seasonal changes, particularly in spring/summer, with a peak in June [53]. Patients who were admitted in the spring or summer (when the ratio of daylight to darkness is longer) had higher rates of IA, earlier onset of illness, longer hospital stays, and admissions for (hypo)manic episodes [54]. IA was also found to be correlated with meteorological factors, such as maximum temperatures and the humidex index (the humidex is a commonly used discomfort indicator of perceived heat arising from the combined effect of excessive humidity and high temperature) [54]. Heat waves due to global climate change are expected to become more frequent and intense and will have negative impacts on health, particularly for those with severe medical and mental health conditions [49,55]. Another Italian study found that there were significantly higher maximum and medium temperatures, as well as humidex, during involuntary admissions compared to voluntary admissions [55]. Studies have shown that heat waves can lead to an increase in hospital admissions for mental problems and emergency room visits [56,57]. Certain drugs like lithium and psychotropics may also alter the effects of heat on the body [58]. 

### 3.3. Impactful Events

Here, nine papers were identified that examined the impact of significant events on IA. Significant development such as economic downturns, changes in cost-sharing policies, the COVID-19 pandemic, instances of public violence, and high-profile court cases connected to IA, were found to have some influence on IAs. 

#### 3.3.1. Economic Downturns and Austerity Measures

Economic recessions can increase the need for mental health care and the treatment gap, with major effects on both individuals and society. A systematic review of 17 studies from different countries found that economic crises were associated with increased prescription drug use and hospital admissions for mental health conditions [59]. A study conducted in Florida found that declining regional economies led to a decrease in community tolerance for individuals seen as a threat, resulting in an increase in involuntary psychiatric hospitalization. This was determined through a time-series analysis of the correlation between unemployment claims and the number of people brought in by the police for examination. The results showed that increased unemployment claims were associated with 309 more male examinations than expected, but there was no association for women [60]. Austerity measures during economic recessions that reduce access to mental health care could also increase IA. In a study investigating the impact of Medicaid cutbacks on state psychiatric hospital utilization (which were typically IA) among people with schizophrenia in the United States, it was found that people who retained Medicaid coverage had little change in involuntary hospitalization, whereas those who lost Medicaid coverage experienced a marked increase in IA over time when other variables were controlled for [61]. Similarly, the rise in IA in the UK has been attributed to three specific impactful events: the economic recession, legislative changes, and the impact of austerity measures on health and social care services [40]. Since the implementation of austerity measures and decreased funding for mental health services and trusts, admission rates for mental health have consistently increased yearly. The trend has risen higher than expected in just two years, indicating the negative impact of austerity measures on mental health services [62]. In the Netherlands, a national reform that increased cost-sharing resulted in a decrease in the utilization of mental health services for both severe and mild problems, particularly in low-income areas. Although this decreased utilization resulted in overall net savings, the reform was linked to expensive increases in involuntary admission and acute mental health treatment for patients with psychotic disorder or bipolar disorder [63]. 

#### 3.3.2. COVID-19 Pandemic 

The COVID-19 pandemic was an unprecedented event that had a significant effect on global healthcare systems. Although the pandemic heightened public awareness of mental health issues, it first appeared to result in a neglect of care for PMHCs. There are conflicting results about the pandemic’s effects in various countries, with some reporting rises in IA during the lockdown and others claiming no changes. While there was a general decrease in the overall number of psychiatric admissions in Portugal, when compared to before the pandemic, the proportion of admissions that were involuntary was greater during the pandemic [64]. In an observational study at a large Swiss emergency department, patients admitted during the COVID-19 pandemic were significantly more likely to be involuntarily hospitalized following their psychiatric consultation [65]. Similarly, a study in the UK during the first lockdown found that, although the overall number of admissions fell by 5.2%, the percentage of IAs increased significantly and steadily. Delayed illness recognition may have led to patients being more susceptible to a decline in insight and missing the window of opportunity for informal admissions, thus being subjected to compulsory admissions [66]. In Italy however, all psychiatric admission rates significantly decreased in the 40 days after the start of the COVID-19 epidemic, but IA admission rates remained unchanged [67]. It was hypothesized that throughout the outbreak, patients and family members may have been more tolerant, avoiding referrals to hospital facilities out of concern for possible exposure to contamination, causing a decline in voluntary admissions [67].

#### 3.3.3. Public Violence

Following violent events, community tolerance for deviant behaviors tends to reduce with a consequent increase in presentations for psychiatric evaluations. Although there have been some widely reported violent attacks committed by individuals displaying overt psychotic symptoms [68], most of the available evidence does not substantiate the notion that mental illness is a major risk factor for violent attacks like mass shootings, and it is instead supported by stigma and public misconception [69,70]. In the weeks following the terrorist attack of September 11, 2001, law enforcement officials in Florida submitted more people for psychiatric examinations to prevent harm to others than would have been predicted by historical patterns [71]. Although it was not reported whether these psychiatric examinations resulted in involuntary admissions, increased demand for IA should be anticipated in planning reactions to future terrorist incidents. 

#### 3.3.4. High-Profile Court Hearings

Other events like high-profile court hearings related to compulsory admissions could trigger a change in the rates of commitments in the following days. There was a sharp increase in IAs immediately after the Cheshire West decision, a high-profile court hearing in 2014 [40]. The decreased informal admissions were attributed to widespread confusion over what constitutes a violation of liberty under the law and concerns that an informal admission would be retrospectively judged as such as in the Cheshire West decision [40]. However, due to the ongoing increases even prior to the Cheshire West decision, the available information was not enough to definitively establish a strong connection between the rise in IA and the court ruling.

### 3.4. Mental Health Legislation 

Six papers investigated how mental health laws and their distinct features affect IA. IA of PMHCs can also be viewed as an indicator of the underlying characteristics of mental health legislation and policies. Since the mid-1970s, several jurisdictions have limited the application of their mental health laws regarding IA to people deemed harmful to themselves or others. This is referred to as the obligatory dangerousness criterion (ODC). The ODC is a requirement for IA in the mental health legislation of many countries, however some other countries permit involuntary treatment without the ODC if the patient is deemed to be unable to consent. Many times, admission must be forced because few patients understand that their symptoms are the result of an illness. The ODC was widely adopted in mental health laws in an effort to achieve a balance between the rights of PMHCs and the need to protect the general population [29]. Patients with first-episode psychosis (FEP) who reside in jurisdictions that use the ODC criterion rather than alternative standards for involuntary treatment would take longer to receive treatment and, as a result, would experience a longer duration of untreated psychosis (DUP) [72]. A recent study found that the mean DUP was on average around five months longer in jurisdictions with an ODC [72]. The extent to which people with unusual social behavior are viewed as a burden depends on the cultural norms and values and could also be reflected in the mental health law [73]. In addition to the ODC, “burden criteria,” such as those found in Swiss mental health law that take into account the burden the patient exerts on family members and other third parties, may also have an impact on IA decision-making. 

Most of the studies relating mental health legislation with IA were pre–post studies that compared the rates of IA before and after the implementation of the mental health legislation. An observational study that assessed the routine data of 15,125 patients admitted to the University Hospital of Psychiatry Zurich between 2008 and 2016 found that the period following the implementation of the new regulation in January 2013 was strongly associated with lower risks of IA [73]. The new law had the goal of reducing stigma and increasing patient autonomy, and the study findings demonstrated that it was indeed connected with a lower risk of IA as no significant changes in mental health care occurred in the catchment area that could have contributed to the disparities in IA risk before and after the new law’s introduction. Similarly, in the 18 years following the implementation of the new mental health law in Italy, the percentage of all admissions that were involuntary decreased significantly [38]. A Chinese study that aimed to look at the rates and correlates of IA after a change in the mental health legislation found that there was a substantial but minor increase in rates of IA among hospitalized patients. Regulations for involuntary placement in the new Chinese mental health law that went into effect in 2013 required that patients demonstrate “self-harm in the immediate past or current risk of self-harm” or “behavior that had harmed others or endangered the safety of others in the immediate past or currently exposed a risk to the safety of others.” The ongoing high prevalence of IAs was attributed to China’s comparatively low number of psychiatric beds per 100,000 people. The legislation did not appear to have reduced IA because of the limited number of inpatient beds in China, cultural stigma, and patient denial of their mental health condition [74]. A Dutch study found a 16% increase in commitments between 2000 and 2004, and a 30% increase in the dangerousness criterion of “arousing hostility”, suggesting a resurgence of paternalism among mental health professionals and a growing intolerance of deviant behavior [75]. 

Few studies have tried to compare IA rates among countries with different commitment criteria and procedures. The study by Salize et al. examined the association between mental health legislation and involuntary admission rates in European countries. They found that having a diagnosed mental health condition is a requirement for detention in every EU member state, with additional criteria such as potential harm to oneself or others being common. There were no significant differences in involuntary admission rates between countries that had the “danger” criterion and those that had the “need for treatment” criterion. Similarly, no significant differences were observed between countries where a non-medical authority—a judge, prosecutor, mayor, or another entity unrelated to the medical system—made the final decision regarding IA and other countries where psychiatrists or other medical specialists made that decision. The study found a tendency towards lower IA rates in countries where a legal representative was involved, which suggests that there are prospects for further investigation [76]. A more recent comparison of annual involuntary hospitalization rates in 22 countries did not find any association with any characteristics of the legal framework [13]. 

### 3.5. Staff Factors 

This category consisted of ten articles that examined staff-related factors including attitudes, personal characteristics such as risk-taking propensity, qualifications, and remuneration, which can potentially impact the decision to commit PMHCs. It is worth noting that most of these studies employed hypothetical scenarios, and it remains uncertain how accurately these scenarios reflect real-life decisions to commit patients. Rather than relying solely on medico-legal criteria and risk assessments, it is apparent that many other factors influence clinicians’ decisions to commit patients involuntarily. Practical considerations such as availability of transportation and lack of outpatient alternative treatments can override medico-legal criteria in making commitment decisions in some cases. Clinicians frequently have to take into account the evaluation setting and patient support networks as well as local resources [41]. In a study to investigate the willingness of resident psychiatrists in North Carolina to commit hypothetical patients who did not fulfil commitment criteria, very few residents said that they would never commit such patients indicating the influence of other contextual variables other than statutory criteria in commitment decisions. Residents weighed the criteria for commitment against the constraints they face in emergency situations, such as the necessity for transportation, a lack of outpatient alternative therapy, and pressure from their superiors [77]. 

#### 3.5.1. Attitudes

There are no consistent findings on which specific traits or attitudes of staff have the greatest influence on their decisions to admit patients involuntarily. Engleman and colleagues found that more experienced US clinicians did not detain more patients but assessed them as being at a higher risk. The clinicians’ detention record in the previous three months was a predictor of their general inclination to detain patients. The authors concluded that the decisions were heavily influenced by the clinicians’ attitudes towards commitment, which were shaped by various factors such as guidelines, previous role models, legal impacts, and personality [41]. A study in Illinois, US, found no link between psychiatrists’ sense of responsibility and their willingness to involuntarily admit PMHCs. However, the study found that psychiatrists were more likely to support hospitalization if they believed in meeting basic needs and less likely if they believed in respecting the right to refuse treatment or the unpredictability of the future [78]. Family pressures and frequent use of involuntary hospitalization were also found to impact the decision to commit a patient. A survey in China showed that one-third of psychiatrists admitted to committing patients against their will due to family pressures [79].

#### 3.5.2. Past Experiences and Risk-Taking Potential

A patient’s perceived danger may be more important than their actual behavior for a psychiatrist to decide whether to seek IA [80]. This perception of danger is influenced by factors such as past experiences. For example, clinicians may become more likely to request involuntary commitment for other suicidal patients if they have previously witnessed a patient’s suicide or if they face malpractice charges for failing to immediately hospitalize patients who later committed suicide [81]. On the other hand, psychiatrists who have been accused of detaining patients against their will unlawfully may be less willing to request involuntary commitment. This leads some psychiatrists to engage in “defensive psychiatry” (defined as the practice of making decisions that limit a psychiatrist’s liability while safeguarding patients and other people from potential harm) [81]. A practicing psychiatrist runs some professional risk when they choose not to pursue an involuntary commitment. Therefore, it seems reasonable to hypothesize that a psychiatrist’s perspective on taking risks affects their decision to seek out involuntary commitment. The findings of a pilot study with psychiatry residents in Massachusetts (US) suggested a significant relationship between an individual psychiatrist’s risk-taking potential and their decision not to seek involuntary commitment. It was assumed that for psychiatrists who had a higher predisposition to accept risks in other areas of their lives, the inherent risks they faced when they chose not to seek involuntary commitment may have been more tolerable [81]. A Swiss study found that more experienced clinicians were less worried about being prosecuted for their actions and having challenges comprehending the legal basis of their actions [82]. 

#### 3.5.3. Qualifications and Knowledge of the Law

Other studies have tried to examine whether specialization in psychiatry could affect rates of IA. Psychiatrists are expected to be more knowledgeable about mental health legislation and more confident in their risk assessments compared to general practitioners, and some countries have restricted the right to commit patients involuntarily to psychiatrists as a measure to reduce the rates of IA. In a Swiss study by Eytan and colleagues, it was found that after the change in policy to make only certified psychiatrists authorized to require compulsory psychiatric admission, those hospitalized were significantly less likely to be hospitalized on a compulsory basis [83]. Concerns of being prosecuted and a lack of knowledge regarding the legal justification for one’s behavior were found to be connected in another Swiss study that sought to understand whether legal considerations varied with physicians’ professional backgrounds and whether they were linked to specific attitudes toward coercion [82]. Understandably, other physicians were more concerned about being indicted for their decisions. Nonetheless, compared to psychiatrists, they disagreed more frequently with the statement that risk assessments of suicide and danger should be exclusive to psychiatrists and agreed more frequently with the statement that a non-psychiatric physicians can assess the risk of suicide and danger [82]. A Chinese study found that physicians with fewer professional titles and lower levels of education were more likely to lack records of the application of risk criterion on the checklists. This was attributed to insufficient systematic training on the IA procedure among those physicians, which may have made them unaware of the reasonable approach for assessing patients’ risks and documenting the relevant circumstances. Psychiatrists who did not complete the risk assessment checklist were older than those who completed it, and this was attributed to elderly psychiatrists having more conservative attitudes and possibly making decisions based on their discretion [84]. A Norwegian study that tried to determine the factors influencing the decisions of general practitioners to commit patients found that half of the GPs worked at public out-of-hour clinics and were less experienced doctors who often did not have any prior relationships with the patients they committed. The GPs also admitted that applying the medico-legal criteria for the referral of involuntary patients was challenging and that they often felt pressured to refer patients [85]. 

In a study of American Psychiatric Association members in the United States, results showed that psychiatrists may not be as knowledgeable about mental health laws as expected. The study found that there was support for limited definitions of mental disorder for involuntary commitment and limited legal grounds for commitment. When asked about involuntary outpatient commitment laws and legal justifications for commitment, many respondents provided incorrect information [86]. Psychiatrists’ support for the various commitment grounds has been reported to be most strongly related to what they perceived the law to be in their state; they tended to support the grounds they thought to be the law. The psychiatrists may have embraced their states’ commitment criteria as their preferences through a process of norm internalization [87]. 

#### 3.5.4. Financial Incentives

Another less explored factor is staff incentives and compensation for involuntary admission orders. Lebenbaum et al. found that an involuntary admission was more likely to occur at emergency departments in Ontario with an increasing likelihood of financial compensation for ordering involuntary admissions [88]. 

### 3.6. Public Attitude and Stigma

This section includes seven papers that explore how negative attitudes and stigma among the general population are connected to support for coercive measures. An important yet frequently overlooked factor that is important in contextualizing IA is the attitude of the public towards PMHCs. Tolerance of deviant behaviors appears to be reducing especially in Western countries, coinciding with a greater emphasis on patient autonomy and rights, as well as strictly defined and regulated coercive measures [89]. The fluctuation of this tolerance in response to impactful events such as economic recession was broached in a previous section of this article. People with mental health problems may exhibit risky behavior that necessitates close observation, sometimes leading to commitment. Their actions could also include, for instance, varying degrees of self-neglect, such as a failure to take proper care of oneself or engaging in other self-destructive behavior that could disrupt family or community life. Although such behavior can be addressed in an outpatient setting without commitment, a low tolerance for such behavior compounded by insufficient alternatives to inpatient care may increase the pressure to admit such people [90]. People in many countries turn to involuntary admission as a result of the absence of sufficient community mental health services and social services. 

#### 3.6.1. Public Attitudes as a Reflection of Mental Health Laws 

Empirical evidence suggests that public attitudes towards coercive measures appear to have a bidirectional relationship with the existing laws on coercive measures. In Norway, the majority of the adult population agreed with the existing laws and regulations governing compulsory admission and treatment in mental health facilities with a stronger societal consensus in favor of coercion based on caregivers’ assessment of the risk of harm. The strongest arguments in favor of involuntary treatment were made when the patient could not care for themselves, such as when they had a depressive illness and suicidal thoughts [91]. In a similar study conducted among French laypeople and medical professionals, 95% of participants agreed that involuntary hospitalization was acceptable in some circumstances, particularly when the patient posed a risk to others in accordance with French law [92]. According to the study, the majority of French laypeople and medical experts agreed that when mentally ill individuals pose a serious risk to themselves and, more significantly, to others, beneficence and public safety must take precedence over patient autonomy [92]. Thus, the laws could be said to be a reflection of public opinion. Similarly, a survey of the acceptance of restrictions on people with mental health problems among the Vietnamese public found that more than three-quarters of the respondents generally accepted compulsory admissions under specific circumstances. This endorsement varied by age and gender reflecting some cultural expectations related to the gender roles in the community [93]. 

#### 3.6.2. Stigma as a Factor in the Endorsement of IA

Other studies have explored the public’s endorsement of restrictive measures and what they perceive to be an appropriate justification for using them. A violent attempt against others was the most endorsed reason for IA in an Austrian study that compared what medical students and journalists thought were legitimate circumstances for compulsory admission. However, about one-third of both groups said that prolonged neglect was a reason. Compared to journalists, the medical students were significantly more likely to endorse commitment and this was attributed to their medical education with a focus on treatment and healing [90]. A Swiss study found that more educated respondents, women, those with a high-degree of social distance towards PMHCs, and those who held unfavorable stereotypes were more likely to endorse IA [94]. Accepting compulsory admissions was found to be predicted by the perceived dangerousness of PMHCs, corresponding to the findings of another Swiss population study in which all three individual compulsory measures (involuntary hospitalization, involuntary medication, and seclusion) were significantly correlated with increased perceptions of danger in PMHCs [95]. These findings corroborate the argument that support for IA by the public could imply support for the custodial component of psychiatry, and thus operationalizing the stigma towards those who have mental health problems [94]. 

A German study that sought to examine the extent to which initiatives for improving the rights of people with mental illness and anti-stigma campaigns were reflected in changes of public attitudes toward restrictions on PMHCs compared data from two population surveys conducted in 1993 and 2011 [96]. There was no change in the level of support for IA but there was a decrease in the opposition to compulsory admission due to reasons not included in the legal criteria in Germany (persecutory delusions, withdrawal from the environment, non-adherence to medication, at the patient’s family’s request, public nuisance). The study findings highlighted the two opposing trends of a general shift towards a more liberal view of patients’ rights and a decreased tolerance and an increased desire for restrictions for inappropriate behavior [96]. The diminished objection to IA for unauthorized reasons was interpreted as a sign of the public’s growing need for protection from those with mental health conditions, as reflected in the results of the 2011 study, which revealed a stronger agreement with the idea that psychiatric hospitals are required to protect society from those with mental health problems [97]. In a similar pattern, a study conducted in the US to assess the evolution of public views on the likelihood of violence from people with mental health problems found that support for IA for those with schizophrenia, depression, and everyday problems increased steadily over the years from 1996 to 2018 [98]. 

There is concern about how the increased fear of those who experience daily life difficulties and the support for their coerced treatment increases the medicalization of daily life crises and social problems with the tendency to use psychiatry and medicine as institutions of social control [99]. However, not all researchers are worried about the increased endorsement of IA among the general public. Lauber et al. argue that acceptance of IA could indicate a good attitude toward psychiatry and an increased level of trust in psychiatry [94]. Following this line of argument, the reduced resistance to IA, for reasons such as persecutory delusions or non-compliance with prescribed medication, could be interpreted as an indication of an increased willingness to seek assistance from these services. The ‘increased trust in psychiatry’ is evident in a recent Swiss study where the belief that psychiatric treatment would be helpful for the fictional vignette character was linked to a greater approval of coercive measures overall, as well as for each specific form of coercive measure studied in accordance with legal requirements [95]. A summary of all the themes and subthemes can be found in Table 2.

### 3.7. Interactions among Contextual Factors Influencing Involuntary Admission 

The number and distribution of mental health and social services is a core factor that is influenced by most other factors influencing IA. The temporal and seasonal associations reported are the result of insufficient service delivery capacity at specific times. Economic recessions that lead to austerity measures including cutbacks in public health spending invariably result in a decrease in service delivery capacity. Economic downturns are also known to reduce tolerance towards deviant behaviors in the community and increase utilization of mental health services causing an increased demand on service delivery capacity that is stretched thin by austerity measures. Another key element with an impact on other factors, and also influenced by other factors, is legislation. The mental health legislation regarding IA reflects the general public’s and mental health professionals’ attitudes toward PMHCs. Conversely, mental health professionals and the general public internalize the prevailing mental health legislation. Public views and attitudes are also important in shaping staff attitudes towards coercive measures, interacting with other factors such as personal traits and practical considerations in deciding to commit PMHCs. The public attitudes are influenced by impactful events that foster stigma such as public violent attacks attributed to mental health problems (Figure 2).

### 3.8. Limitations and Biases of Selected Studies

The studies included in the review have several limitations that impact their validity and generalizability. First, there are limitations related to study design and methodology. For many reasons including ethical ones, none of the studies conducted in this field were RCTs. Many studies relied on retrospective data collection, introducing potential biases and incomplete information. Moreover, the use of administrative databases may have resulted in an incomplete understanding of the variables of interest, and the absence of standardized assessment instruments affects comparability and measurement validity. The heterogeneity of data gathered across various services, jurisdictions, and countries adds a significant level of complexity when attempting to make comparisons. Additionally, incomplete data on patients’ admission history and illness profiles restrict a comprehensive understanding of the factors influencing outcomes. Inaccuracies and lack of standardization in diagnosis information impact the reliability of findings. The fact that some studies were cross-sectional hampers the ability to establish causal relationships and track changes over time. Studies with ecological study designs may have overlooked individual-level nuances. Furthermore, there are limitations regarding generalizability and the samples. The studies often focused on specific services, catchment areas, or countries, limiting the applicability of findings to different contexts. Unclear definitions of the base population also make it difficult to determine the relevance of the results. Some studies failed to measure all relevant variables, leading to potential confounding and incomplete understanding of the factor under investigation. Subjective information from questionnaires introduces social desirability bias, affecting response accuracy. Future research should aim for more rigorous study designs, standardized assessment instruments, and comprehensive data collection methods to enhance the validity and applicability of findings in the field.

## 4. Discussion

This review aimed to identify and synthesize the literature on non-patient-related factors thahave the potential to influence IA. The paper identifies factors that contextualize IA in various settings, grouped under categories of service characteristics, mental health legislation, impactful recent events, temporal and seasonal variables, staff variables, and public attitudes and stigma. It also reveals interrelationships among these factors. The availability and distribution of mental health and social services significantly influence IA, with insufficient service capacity during certain periods leading to temporal and seasonal associations. Economic downturns and austerity measures reduce service delivery capacity, and also increase community intolerance towards deviant behaviors, further straining mental health services. Mental health legislation reflects public and professional attitudes towards psychiatric patients, and conversely, public attitudes and perceptions are shaped by stigma-inducing events like violent attacks tied to mental health conditions. These factors, in conjunction with the personal traits of mental health professionals and practical considerations such as the availability of alternatives to in-patient care, interact in the decision-making process concerning IA. The relationships mostly center on the need for services and the ability to meet those needs. A few studies acknowledge the essential connection among the factors being examined, while others fail to do so. For example, the investigation to test the hypothesis that the contraction of regional economies affects the incidence of involuntary admissions to psychiatric emergency services focuses on reduced societal tolerance and fails to acknowledge the possible decline in access to mental health services as a result of the economic recession [60]. The study investigating the consequences of austerity measures in the UK also overlooks the potential influence of decreased societal tolerance towards deviant behaviors during the economic recession [40]. Similarly, the studies on the influence of staff-related factors on the decision to involuntarily admit only assume the impact of personality traits and existing laws on staff attitudes [81,82,87] and fail to consider how the prevailing public attitudes and stigma may have shaped those attitudes.

This review’s assessment of various contextual factors that might affect IA is not exhaustive. Other factors such as poverty and inequalities, economic indices such as GDP, budgetary allocations for mental health care, number of forensic psychiatric services, and ethnic density have been linked with IA [13,18]. However, it draws attention to the reality that IA often happens in the context of complex interwoven factors that are often not solely related to the patient or the mental health professionals. While it may be difficult to estimate the magnitude of the effect of these contextual factors, they seem to contribute significantly to the problem and could present opportunities for long-term sustainable interventions to reduce the need for IA. These contextual factors do not act in isolation but rather interact and reinforce each other, so strategies to reduce IAs have to be robust enough to address these influences by employing multimodal approaches. Given how other factors in the society are implicitly involved, it also appears that the task of reducing IA should not be left to only mental health professionals.

The existing laws concerning mental health care are crucial in shaping various contextual factors. Both the general public and psychiatrists internalize these laws, and their attitudes and beliefs towards involuntary admission are significantly influenced by them. Therefore, altering the laws and policies related to involuntary admission is essential in shaping psychiatrists’ values and practices, possibly leading to a reduction in involuntary admission. However, legislation reform alone will not suffice to address the problem without the optimal reform of services and training of the staff required to implement the laws. A well-developed and responsive mental health care system will result in consistent high-quality service delivery, preventing situations that lead to IA. It is evident that even when the legislation is very protective of the rights of PMHCs, other failings of the public health and social systems create situations that make IAs inevitable. Since nonclinical factors such as specific attitudes and previous experiences may influence clinicians’ decisions to request involuntary commitment, perhaps generic education and training may not be enough to cause the needed change in practice and attitudes. Specific targeted interventions, particularly for residents at the beginning of their training, may be useful to reduce the influence of these nonclinical factors such as through peer assessment, monitoring, and, in certain cases, individual counseling and psychotherapy [81]. 

The seasonal and temporal trends in the rates of IA imply that hospitals can anticipate quantifiable pressure using models that incorporate regular temporal and seasonal patterns. With the sophisticated analytical methods and prediction models employed in healthcare analysis and forecasting, it is possible to predict periods of increased demand for emergency care and plan appropriately since an overwhelming ratio of patients to staff is a frequently cited reason for IA and other coercive measures. This is particularly useful in optimizing staffing schedules and informing resource utilization in resource-limited settings. It is also suggested that psychiatrists closely monitor or modify therapy for patients that are sensitive to weather changes [100].

These study findings on the impact of economic downturns suggest that contracting economies may signal the need for greater capacity to meet increased demand for mental health care and avoid the worsening of symptoms in vulnerable populations. In the long run, austerity measures that cut back on funding for mental health care are costly since they invariably lead to an increase in the number of IAs. The estimation and prediction of demand for mental health services and variations in a society’s tolerance for deviant behavior during economic downturns can assist healthcare managers in planning their services. Relevant authorities have the opportunity to develop measures to mitigate the negative effects of declining economies on mental health. For example, providing preventive mental health care instead of IA may be beneficial. Adapting techniques used to forecast short-term changes in employment to predict the need for psychiatric services and optimize resources is recommended for further investigation [60]. Robust interventions that incorporate resilience mechanisms for mental health systems during adverse events such as economic recessions or the COVID-19 pandemic are particularly needed. During times of communal anxiety, police and care providers may well be made aware of, and cautioned against, diminished tolerance for those with mental health conditions [71]. Sentinel surveillance of emergency psychiatric services could be useful, and those who monitor these services may want to look into other possible causes of unusually low or high usage. It is important to note that this will only be feasible in services and countries with the infrastructure required to maintain detailed uniform data about IAs.

All stakeholders involved in IA, including family, community members, police, mental health personnel, judiciary staff, legislators, and the individual with the mental health condition, are products of society and, to some extent, are shaped by prevailing norms and societal values. Given the evolution of public attitudes and stigma, it is important to monitor cultural beliefs, attitudes, and tendencies that link mental health conditions with violent behavior. Such beliefs are still prevalent, and the public may equate them with involuntary treatment, resulting in formal public health regulations that significantly restrict the liberties of PMHCs [98]. The relationship between mental health legislation, stigma, and endorsement of the restriction of liberties of PMHCs [87,92,94,95] suggests that stigma reduction among the general public is fundamental in addressing the problem of IA. Perhaps a “fusion law,” as proposed by Dawson and Szmukler [101], that eliminates the exclusivity to PMHCs laws that restrict liberties based on impaired decision-making capacity and not on a mental disorder diagnosis, could be less stigmatizing and useful in reversing the rising trend of endorsement of IA for PMHCs.

Evaluating national and region-wide policies relies on internationally standardized and yearly updated IA rates (detailing several variables such as regular or emergency admission as well as socio-demographic and clinical characteristics) [76]. It is not surprising that the majority of the research on the topic originates from high-income countries with seemingly high rates of IA, which is possibly because these countries have maintained reliable statistics on IA for a long time. Keeping reliable data on IA rates at local and national levels is possibly the first step in evaluating the problem and developing interventions to address it. Improving existing international standards for reporting on mental health is necessary to ensure the validity of future investigations and serve as a foundation for additional in-depth studies in the area. Other contextual factors that might affect access to mental health care and cause mental health conditions to worsen to the point where IA is required, such as the level of universal health care coverage and policies related to health care for immigrants, particularly in Western countries, have not been investigated. Out-of-pocket spending and a lack of legal framework are significant barriers to accessing mental health care in many developing countries, but few studies have explored how this affects IA in these contexts. Additionally, a country’s level of respect for the human rights of the general population can be used to estimate how well the rights of PMHCs are protected. 

### 4.1. Strengths and Limitations

This review’s broad inclusion criteria, which limit quantitative synthesis, can be considered a strength because they allow us to see how different factors interact to influence IA. Although some studies did not establish a direct link between contextual factors and involuntary admission (IA), we included them in our analysis to provide a broader perspective on the issue. Despite our conviction that a narrative review is more suitable for analyzing a wide range of factors, such as those discussed in this paper, it is important to note that narrative reviews allow authors a lot of flexibility in interpretation, which increases the likelihood of bias. Nevertheless, we have taken steps to reduce the potential for bias by concentrating on quantitative studies published in peer-reviewed journals. The exclusion of qualitative studies and studies published in other languages limits the scope of contextual factors explored in this article. 

### 4.2. Future Directions

There is a gap in the literature in terms of studies that explore how the level of universal health coverage and universal human rights indices affect IA rates. Future temporal studies can explore the impacts of periods such as holidays while controlling for other variables such as general medical problems and poor treatment adherence, which may also contribute to the beginning of an acute clinical picture. Studies examining staff factors and public attitudes typically do not investigate their direct impact on IA. Therefore, it would be compelling to explore the potential direct impact they may have on IA. The impact of environmental factors on IA needs to be further investigated in light of the growing effects of climate change. In countries with public acts of violence like mass shootings in the United States, it would be interesting to examine the shifts in public attitudes towards PMHCs and approval of restrictions of liberties. It is also important to keep an eye on how other policies, such as those pertaining to gun control, may affect these attitudes and legislation pertaining to IA. Given that a small number of countries have conducted the majority of the research on the subject, it is crucial that other countries look into the contextual factors that are relevant to their setting. There is a need to better comprehend the complex interaction between austerity measures, policies, and IA. Considering how policymakers are interested in cost-saving interventions, it is particularly important to include economic outcomes in studies that investigate the effectiveness of alternatives to IA. The variety of contextual factors also emphasizes the need for multidisciplinary collaboration between healthcare researchers and researchers from other fields, such as the social sciences, to build sustainable strategies. Due to the diversity of contextual factors that may influence IA, it is apparent that some factors have been left out and so there is need a for more review articles, especially those including qualitative studies. 

## 5. Conclusions

This paper has attempted to contextualize factors that contribute to the use of IA. The findings indicate that various interacting factors are at play, beyond lawful criteria for IA. The absence of a well-developed mental health system, impactful events such as economic recessions, mental health legislations, seasonal and temporal factors, staff factors, and public attitudes are some of the contextual factors that can influence IA. They present opportunities for multimodal approaches to developing interventions that reduce IA. The complexity of the factors influencing IA and the implicit role of society as a whole also suggest that resolving it will require a social change.

## Figures and Tables

**Figure 1 healthcare-11-01986-f001:**
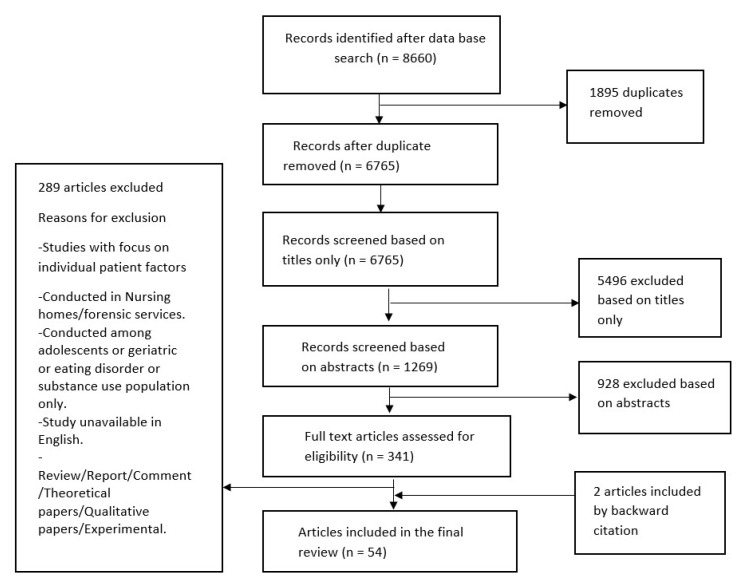
Flow chart of articles included and excluded after the review.

**Figure 2 healthcare-11-01986-f002:**
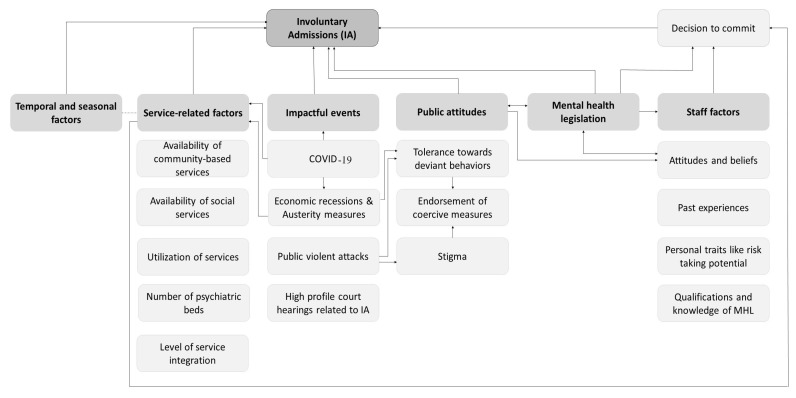
Interactions among contextual factors influencing involuntary admission.

**Table 1 healthcare-11-01986-t001:** Inclusion and exclusion criteria.

Inclusion criteria	Peer-reviewed published full-text articles available in English that met the following criteria: Explicit indication of association between involuntary admission or the decision to commit and factors(s) apart from those related to the patients. Observational studies in general psychiatry for adults. Studies that have explored the opinions of the general public about IA.
Exclusion criteria	Excluded if one or more of the following criteria are met: Studies conducted exclusively among patients with eating disorders, dementia, or intellectual disabilities, forensic patients, adolescent or geriatric patients, or people with substance use disorders only. Studies that explored factors related to patients treated involuntarily in the community or nursing homes rather than in a hospital setting were also excluded. Qualitative studies, reviews, editorials, and theoretical papers Studies testing an intervention to reduce IA

**Table 2 healthcare-11-01986-t002:** A summary table of derived themes and subthemes.

S/N	Themes and Sub-Themes
1	**Service-related factors**
	Availability of community-based services [28,29]
	Efficiency of services [31,33,37].
	Availability of complementary social services [35,36,37]
	The number and location of psychiatric hospital beds [13,38,39,40,42,43,44]
2	**Temporal and Seasonal factors**
	Temporal [36,45,46,47,48,49].
	Seasonal [53,54,55].
3	**Impactful events**
	Economic downturns and austerity measures [40,60,61,63]
	COVID-19 pandemic [64,65,66,67].
	Public violence [71]
	High-profile court hearings [40]
4	**Mental health legislation** [13,38,40,73,74,76]
5	**Staff factors**
	Attitudes and beliefs [41,78]
	Past experiences and personal traits [81,82]
	Qualifications and knowledge of MHL [77,83,84,85,86,87]
	Financial incentives [88]
6	**Public Attitudes**
	Public attitudes as a reflection of mental health laws [91,92,93]
	Stigma as a factor in the endorsement of IA [90,94,95,97]

## Data Availability

A summary table with information about the selected studies can be found in the Appendix A.

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
