# Peer review of "Beyond Patient Characteristics: A Narrative Review of Contextual Factors Influencing Involuntary Admissions in Mental Health Care"

_healthcare, 2023, doi:10.3390/healthcare11141986_

Round 1

Reviewer 1 Report

This study aims at investigating contextual factors that can impact IA, through a narrative review of papers that have empirically tested these associations. I think that the topic is relevant and that the paper can give an important contribute to the field. However, I think that many shortcomings undermine the quality of the review and should be addressed before the paper being suitable for publication.

RESULTS

The authors stated that they have followed SANRA scale for the quality of the narrative review. However, it does not seem that all the requirements were met. In the following, I will list my concerns based on items in that scale:

Item 2. While the aims seem to be well specified and consistent with most of the results section, the last part of the results does not seem to fit very well with the rest. Up to that part, the authors reviewed papers that tested empirically the impact or the relation between IA and some factors. However, the section “Public attitude & stigma” collect papers that did not show empirically relations between factors and IA. The authors acknowledge this difference in the conclusion, but I think this is not very appropriate, and should be either anticipate in the aims (as a separate aim, but then the authors should be careful with the search strategy) or should be removed.

Item 3 is not completely met. The description of the search strategy is quite approximate. Which exact terms has been used? Which ones are excluded? Was there a specific population? Also, you said that you want to focus only on contextual factors, but your search strategy did not reflect that. What were the “relevant” criteria to reduce the papers from 8660 to 56? Please clarify all these points. Maybe using a PICO can help to make these points clearer.

Regarding item 5 of SANRA, although the reviewed papers are described quite carefully and in a functional way to present a scientific point, the evaluation of the quality of evidence (for instance, from RCT, or cross sectional design) is missing. Please review carefully that through the result section. It can help to have a table with the description of all the studies included in the review, for instance, where all these details can be presented.

Item 6. Appropriate data are presented sometimes but not in a systematic way. For instance, the authors presented odd ratios sometimes to describe the impact of a certain factor, but often they did not cite any indicator.

Also, in the presentation of the results, there are some parts that belong more to the discussion than results (for instance p. 3 lines 115-123, p. 11 lines 501-516).

Discussion:

The authors stated that the review is not exhaustive and that other factors have been linked to IA. I wonder why if it is so, the papers where these other factors were tested were not included in the present review. Please provide a rationale for that.

Author Response

We thank you for taking the time to review our manuscript and providing detailed and constructive feedback. Your comments helped us to improve our manuscript. To address all the comments, relevant texts, tables and diagram have been included. The new version of the manuscript has all the added text in tracked changes. The response is point-by-point, with excerpts from the manuscript in some cases. We hope to have addressed all your concerns.

Review comments

This study aims at investigating contextual factors that can impact IA, through a narrative review of papers that have empirically tested these associations. I think that the topic is relevant and that the paper can give an important contribute to the field. However, I think that many shortcomings undermine the quality of the review and should be addressed before the paper being suitable for publication.

RESULTS

The authors stated that they have followed SANRA scale for the quality of the narrative review. However, it does not seem that all the requirements were met. In the following, I will list my concerns based on items in that scale:

Item 2. While the aims seem to be well specified and consistent with most of the results section, the last part of the results does not seem to fit very well with the rest. Up to that part, the authors reviewed papers that tested empirically the impact or the relation between IA and some factors. However, the section “Public attitude & stigma” collect papers that did not show empirically relations between factors and IA. The authors acknowledge this difference in the conclusion, but I think this is not very appropriate, and should be either anticipate in the aims (as a separate aim, but then the authors should be careful with the search strategy) or should be removed.

Response: While we recognize the importance and relevance of this comment, we feel that that public attitude is an important yet often overlooked factor to be considered. We have tried to explain this discrepancy earlier in the manuscript as you have suggested.

Although no research specifically investigated the direct link between public attitudes towards IA and IA rates, we recognized that this aspect is frequently disregarded despite its substantial influence on other contextual factors. Consequently, we chose to incorporate the existing studies that have examined public opinions on IA, even if they do not explicitly demonstrate their impact on IA rates.”

Item 3 is not completely met. The description of the search strategy is quite approximate. Which exact terms has been used? Which ones are excluded? Was there a specific population? Also, you said that you want to focus only on contextual factors, but your search strategy did not reflect that. What were the “relevant” criteria to reduce the papers from 8660 to 56? Please clarify all these points. Maybe using a PICO can help to make these points clearer.

Response: We greatly appreciate this suggestion as it helped us identify an error in our manuscript where two studies evaluating two contextual factors were inadvertently counted twice. We have now rectified this mistake throughout the entire document.  We have also broadened the methods section to include sub-sections for more details. We have included a table of inclusion and exclusion criteria (Table 1) and a PRISMA flow chart showing the study selection process (Figure1).

Regarding item 5 of SANRA, although the reviewed papers are described quite carefully and in a functional way to present a scientific point, the evaluation of the quality of evidence (for instance, from RCT, or cross-sectional design) is missing. Please review carefully that through the result section. It can help to have a table with the description of all the studies included in the review, for instance, where all these details can be presented.

Response: We submitted the manuscript with a table including the summary of the study findings attached as supplementary material. Unfortunately, it seems this table was not shared with the reviewers. We have now included the table in the manuscript and added a section with the study design as suggested.

Item 6. Appropriate data are presented sometimes but not in a systematic way. For instance, the authors presented odd ratios sometimes to describe the impact of a certain factor, but often they did not cite any indicator.

Response: We have tried to make the data presented more uniform, although this is difficult since the studies have employed different statistical methods to assess these contextual factors. We highlight this heterogeneity of results as a limitation to the studies selected.

Also, in the presentation of the results, there are some parts that belong more to the discussion than results (for instance p. 3 lines 115-123, p. 11 lines 501-516).

Response: We tried to provide some brief background information on each contextual factor before presenting the result to facilitate understanding.

Discussion:

The authors stated that the review is not exhaustive and that other factors have been linked to IA. I wonder why if it is so, the papers where these other factors were tested were not included in the present review. Please provide a rationale for that.

Response: The impact of ethnic variations on IA has been published as a systematic review.  Also, as we stated in the limitations, qualitative studies and studies not available in English were not included.

We thank the reviewer once again for their helpful suggestions which have improved the quality of the manuscript.

Reviewer 2 Report

To authors, 

I think this study will be a good basis for future policy proposals on IA.  

My comment can be taken as a suggestion.

Methods section: 

1) Please present your research question with more emphasis.

2) Present a flow diagram representing the study selection process.

3) Add a data collection period.

Other comments:

1) The first time an acronym appears, include the full term to help readers with literacy (e.g., PMHC).

2) I think the results would be clearer if you could number the subsystems of the themes you identified or tabulate the resulting categories.

3) It would be nice if the list of the 56 final studies included were presented separately from the references. 

I wish you all the best in your revision work and hope to see this manuscript as a published research article.

Sincerely, your reviewer

Author Response

We thank you for taking the time to review our manuscript and providing detailed and constructive feedback. Your comments helped us to improve our manuscript. To address all the comments, relevant texts, tables and diagram have been included. The new version of the manuscript has all the added text in tracked changes. The response is point-by-point, with excerpts from the manuscript in some cases. We hope to have addressed all your concerns.

Reviewer 2

I think this study will be a good basis for future policy proposals on IA.  

My comment can be taken as a suggestion.

Methods section: 

1) Please present your research question with more emphasis.

Response: Thank you for pointing this out. We have specified this in the introduction.

“This review aims to fill this gap by including empirical observational studies to provide a broad overview of factors, beyond those related to the patient, that could influence IA.”

“The objective of this paper is to identify and synthesize literature on non-patient-related factors that have the potential to influence IA.”

2) Present a flow diagram representing the study selection process.

Response: We greatly appreciate this suggestion, as it helped us identify an error in our manuscript where two studies evaluating two contextual factors were inadvertently included twice. We have now rectified this mistake throughout the entire document.   We have included a PRISMA flow diagram in the methods section

3) Add a data collection period.

Response: We have included the date in the methods section.

“Four accredited electronic databases (PubMed, PsychINFO, EMBASE and Web of Science) were iteratively searched between 20th July 2022 and 11th August 2022 using specific search strings”

Other comments:

1) The first time an acronym appears, include the full term to help readers with literacy (e.g., PMHC).

Response: We apologize for this oversight. We have included the full term at the beginning.

2) I think the results would be clearer if you could number the subsystems of the themes, you identified or tabulate the resulting categories.

Response: We have numbered the subheadings as suggested

3) It would be nice if the list of the 56 final studies included were presented separately from the references. 

Response: We did include a table with a summary of the included studies as a supplementary material but it was not shared with you reviewers. We have now included it within the manuscript.

I wish you all the best in your revision work and hope to see this manuscript as a published research article.

Sincerely, your reviewer

We thank the reviewer once again for their helpful suggestions which have improved the quality of the manuscript.

Reviewer 3 Report

I read the paper “ Beyond patient characteristics: A narrative review of contextual factors influencing Involuntary Admissions in mental health care”. The authors have adequately described the background and research objectives of their review but the paper overall demonstrates significant methodological weaknesses. I believe that the paper requires significant re-writing to be suitable for publication

See main points below:

1)      The authors explain that they conducted a narrative review but then they noted that “all quantitative studies were included” (line 84). Were qualitative or mixed methods studies excluded? Considering that the authors were specifically interested in capturing contextual variables, I was expecting to see these types of studies included in their synthesis. Alternatively, they need to provide a clear rationale for not including them.

2)      In line 99 the authors note that “various contextual factors were identified”. How did that identification take place? The only explanation provided is in lines 102-104 (“When a study included patient-related factors as well as a contextual factor of interest, only the contextual factor was considered). As the authors included quantitative studies it is important to clarify how were those contextual factors assessed in the primary studies and how the authors define a “contextual” factor.

3)      It is unclear how the review process was conducted. For example, what type of data was extracted and why? Did the authors use any specific extraction forms (if yes, this should be included in the appendix)

4)      The authors do not provide any tables showing what context variables were identified in each paper and relevant information about the type of study, population, measures used etc

5)      The authors do not describe what processes they followed to construct the themes reflecting the contextual factors that were addressed in the included studies.

6)      The authors note in lines 73-74 that “The SANRA scale for the evaluation of the quality of narrative review articles informed the methodological approach of this narrative review, and then in lines 104-106 that “The review was intended to encompass a wide range of literature on the subject, and no quality rating of the articles was performed as it is considered optional for narrative reviews”.  However, the authors do not offer a separate section discussing the methodological quality of the included studies. Even if a detailed rating table is not included, they still need to provide a comprehensive evaluation of the methodological quality of the included studies and sources of bias that may reduce their quality.

Author Response

We thank you for taking the time to review our manuscript and providing detailed and constructive feedback. Your comments helped us to improve our manuscript. To address all the comments, relevant texts, tables and diagram have been included. The new version of the manuscript has all the added text in tracked changes. The response is point-by-point, with excerpts from the manuscript in some cases. We hope to have addressed all your concerns.

Review comments

I read the paper “ Beyond patient characteristics: A narrative review of contextual factors influencing Involuntary Admissions in mental health care”. The authors have adequately described the background and research objectives of their review but the paper overall demonstrates significant methodological weaknesses. I believe that the paper requires significant re-writing to be suitable for publication

See main points below:

1)      The authors explain that they conducted a narrative review but then they noted that “all quantitative studies were included” (line 84). Were qualitative or mixed methods studies excluded? Considering that the authors were specifically interested in capturing contextual variables, I was expecting to see these types of studies included in their synthesis. Alternatively, they need to provide a clear rationale for not including them.

Response: While we recognize the relevance and importance of this comment, we chose not include qualitative studies mostly because they were mostly focused on facilitators and barriers to implementing interventions. We have clarified that qualitative studies were excluded and have tried to give our reasons in the methods section and we highlight it as a limitation to our review at the end  of the article.

We chose to exclude qualitative studies because majority of the qualitative studies that assessed the factors of interest were related to the implementation of an intervention or subjective opinions of professionals, service users and families of service user.”

2)      In line 99 the authors note that “various contextual factors were identified”. How did that identification take place? The only explanation provided is in lines 102-104 (“When a study included patient-related factors as well as a contextual factor of interest, only the contextual factor was considered). As the authors included quantitative studies it is important to clarify how were those contextual factors assessed in the primary studies and how the authors define a “contextual” factor.

Response: We greatly appreciate this suggestion, as it helped us identify an error in our manuscript where two studies evaluating two contextual factors were inadvertently counted twice. We have now rectified this mistake throughout the entire document.  We have explained what we refer to as a contextual factor in our study.

In healthcare literature, contextual factors are often viewed as barriers or facilitators to implementing an intervention [23].  For the purpose of this review, we define contextual factors as diverse elements within a specific setting that are unrelated to the patients themselves but can exert a significant influence on IA directly or indirectly while IA refers to compulsory admission under a mental health act.

We have also created a subsection on the selection process, and included a PRISMA flow chart to elaborate more. We also included a table with a summary of each selected study. (Figure 1)

3)      It is unclear how the review process was conducted. For example, what type of data was extracted and why? Did the authors use any specific extraction forms (if yes, this should be included in the appendix)

Response: We added a subsection on the data abstraction process and included a summary table of selected studies within the manuscript, rather than as supplementary material in the initial submission. (Table 2)

4)      The authors do not provide any tables showing what context variables were identified in each paper and relevant information about the type of study, population, measures used etc

Response: We have now included a summary table of the included studies in the results section. (Table 2)

5)      The authors do not describe what processes they followed to construct the themes reflecting the contextual factors that were addressed in the included studies.

Response: We have tried to explain how the themes were formed in the methods section.

We employed a combination of deductive reasoning, utilizing factors we hypothesized, and inductive reasoning, based on our analysis of the selected papers, to establish categories of contextual factors. These categories served as a framework for organizing the papers accordingly.”

6)      The authors note in lines 73-74 that “The SANRA scale for the evaluation of the quality of narrative review articles informed the methodological approach of this narrative review, and then in lines 104-106 that “The review was intended to encompass a wide range of literature on the subject, and no quality rating of the articles was performed as it is considered optional for narrative reviews”.  However, the authors do not offer a separate section discussing the methodological quality of the included studies. Even if a detailed rating table is not included, they still need to provide a comprehensive evaluation of the methodological quality of the included studies and sources of bias that may reduce their quality.

Response: Thank you for the comment, we have included a subsection about the limitations and biases in the included studies.

We thank the reviewer once again for their helpful suggestions which have improved the quality of the manuscript.

Round 2

Reviewer 1 Report

The authors did a good job in addressing all the comments. I do not have additional comments.

Author Response

We thank the reviewer for their comments which have greatly improved the quality of the manuscript.

Reviewer 3 Report

Thank you for your revised paper “Beyond patient characteristics: A narrative review of contextual factors influencing Involuntary Admissions in mental health care “.

The paper has significantly improved and addressed many of my concerns regarding its methodology. Furthermore, their discussion section offers good insights on the value of their research in policy-making planning.

There are though a few remaining issues that need to be addressed:

1)      The authors have added correctly a table presenting the key characteristics of the studies that were included in their review and later on they present a model showing the effects of different contextual factors in Involuntary admission. However, the presentation of the findings is a bit confusing because there is no reference and explanation about the figure’s subthemes earlier on finding sections. I had to constantly go back and forth between the main table and the sub-sections to try to understand how and why those sub-themes were created.

I recommend the authors add a table with all the sub-groups or sub-themes that belong to each of their main themes and mention the studies that are relevant to. 

2)      In the beginning of the discussion chapter the authors need to re-iterate the research objective of the study. They also need to add a short reference to their findings overall and the relationships that emerged between the factors and the sub-factors that feature in figure 1. Then they need to discuss those within the context of relevant literature. For example, how have service characteristics been studied before or how have the relationships shown between the  sub-factors with the study’s figure  been addressed in other studies and what are the similarities and differences in comparison to the authors' research findings.
